# Characterisation of Bacteriophage vB_SmaM_Ps15 Infective to *Stenotrophomonas maltophilia* Clinical Ocular Isolates

**DOI:** 10.3390/v14040709

**Published:** 2022-03-29

**Authors:** Dragica Damnjanović, Xabier Vázquez-Campos, Lisa Elliott, Mark Willcox, Wallace J. Bridge

**Affiliations:** 1School of Biotechnology and Biomolecular Sciences, The University of New South Wales, High Street, Kensington, NSW 2052, Australia; 2NSW Systems Biology Initiative, School of Biotechnology and Biomolecular Sciences, The University of New South Wales, High Street, Kensington, NSW 2052, Australia; x.vazquezcampos@unsw.edu.au; 3AusPhage, 21 Everett Street, Mount St John, QLD 4818, Australia; ausphage@gmail.com; 4School of Optometry and Vision Science, The University of New South Wales, High Street, Kensington, NSW 2052, Australia; m.willcox@unsw.edu.au

**Keywords:** *Stenotrophomonas maltophilia*, bacteriophage, *Menderavirus*

## Abstract

Recent acknowledgment that multidrug resistant *Stenotrophomonas maltophilia* strains can cause severe infections has led to increasing global interest in addressing its pathogenicity. While being primarily associated with hospital-acquired respiratory tract infections, this bacterial species is also relevant to ophthalmology, particularly to contact lens-related diseases. In the current study, the capacity of *Stenotrophomonas* phage vB_SmaM_Ps15 to infect ocular *S. maltophilia* strains was investigated to explore its future potential as a phage therapeutic. The phage proved to be lytic to a range of clinical isolates collected in Australia from eye swabs, contact lenses and contact lens cases that had previously shown to be resistant to several antibiotics and multipurpose contact lenses disinfectant solutions. Morphological analysis by transmission electron microscopy placed the phage into the *Myoviridae* family. Its genome size was 161,350 bp with a G + C content of 54.2%, containing 276 putative protein-encoding genes and 24 tRNAs. A detailed comparative genomic analysis positioned vB_SmaM_Ps15 as a new species of the *Menderavirus* genus, which currently contains six very similar globally distributed members. It was confirmed as a virulent phage, free of known lysogenic and pathogenicity determinants, which supports its potential use for the treatment of *S. maltophilia* eye infections.

## 1. Introduction

*Stenotrophomonas maltophilia* is an aerobic, Gram-negative bacillus, with a remarkable ability to adapt to various environmental conditions [1]. Its most common natural habitats are plant rhizospheres, where they may protect plants from bacterial and fungal pathogens [1,2]. *S*. *maltophilia* have also been recovered from fresh water and wastewater [2], animals [3] and foods, such as raw milk and cheese [4]. In addition to existing as an environmental commensal, *S*. *maltophilia* has been implicated as an aetiological agent of various infectious diseases (reviewed in [5]) including community-acquired infections in children and adults [2,6].

*S. maltophilia* had been frequently isolated from patients with chronic lung diseases, such as cystic fibrosis, but was previously regarded as a common coloniser of the respiratory tract of severely immuno-compromised hosts rather than an invasive pathogen [7]. It has since been recognised as the third most common opportunistic pathogen in hospitals following *Pseudomonas aeruginosa* and *Acinetobacter*
*baumannii* [8] and shown to make a significant contribution as a constituent of polymicrobial infections associated with increased mortality in pneumonia patients [9]. Its natural resistance to a range of antibiotic classes and its ability to develop resistance to new antimicrobials via multiple molecular mechanisms has warranted its inclusion on the global priority list of the TOp TEn resistant Microorganisms (TOTEM) as a medium priority pathogen [10].

Microbial keratitis is a serious eye disease that can lead to blindness [11]. *P. aeruginosa* is the most common Gram-negative isolate from microbial keratitis associated with contact lens wear [11]. However, *S. maltophilia*, which is a common contaminant and adherent of contact lenses and lens cases [12,13] has also been shown to cause ocular disease [14,15]. A recent study has reported that the mean duration of treatment for *S. maltophilia* keratitis was approximately twice as long (10 days) as for *P. aeruginosa* keratitis (4 days) [16]. Additionally, it has been suggested that *S. maltophilia* can promote the propagation of *Acanthamoeba* in contact lens cases leading to the subsequent amoebic infection of the cornea [17]. This is due to its capacity to live and multiply inside *Acanthamoeba* spp. [18]. Such endocytobiotic relationships can upregulate virulent genes of intracellular bacteria and enable their survival and transmission upon amoebae lysis or during secretion via amoebae vesicles, while also increasing amoebal pathogenicity and resistance to biocides [19,20].

Phage therapy is one strategy for treating bacterial infections that is increasingly gaining interest, particularly considering rising antibiotic resistance. However, the research exploring phage therapy candidates for application in the ocular field has been rare. In the 1970s and 1980s, Eastern European researchers reported successful treatment of conjunctivitis and blepharitis with antistaphylococcal phages in the form of eye drops, in addition to full recovery from suppurative eye infections [21]. More recently, studies in animal models have yielded promising results with bacteriophage eye drops, improving *P. aeruginosa* keratitis [22,23]. Intravitreously administered phages for vancomycin-sensitive and resistant enterococcal endophthalmitis has shown to be effective in mice with no toxic effect on retinal function [24]. The successful in vitro elimination of *Staphylococcus* spp. isolated from dogs with bacterial conjunctivitis led to a patented method for the preparation of stable eye drops formulations of phage cocktails for in vivo animal treatments [25]. One research group has translated phage therapy to humans for the treatment of long-term vancomycin resistant *S. aureus* keratitis [26].

The majority of *S. maltophilia* phages isolated to date belong to the virulent *Siphoviridae* family, followed by *Myoviridae* and *Podoviridae,* with a smaller number of temperate and filamentous phages (reviewed in [27]). While some of these phages have been screened for lytic activity against clinical isolates without specifying the infectious source [28,29,30], there are no reports to date of phages that are infective to human ocular *S. maltophilia* isolates. Therefore, the aim of this study was to explore the lytic potential of the phage vb_SmaM_Ps15 against ocular isolates of *S. maltophilia* with the long-term possibility to be used to treat eye infections.

## 2. Materials and Methods

### 2.1. Bacterial Strains

Host bacterium *P. aeruginosa* AP143 was provided by AusPhage as an “ex PAO1” strain. However, a subsequent examination has revealed that the AP143 used for phage propagation was a mixed strain, consisting of two species, *S. maltophilia* and *P. aeruginosa*. The colonies of each species were purified by streaking three times on Tryptone Soya Agar (TSA) (CM0131; Oxoid Australia, Thebarton, Australia), followed by propagation in Tryptone Soy Broth (TSB) (CM0129; Oxoid Australia) at 37 °C for 24 h without shaking, and stocking as AP143S and AP143P, respectively. The following experiments were conducted using AP143S.

Clinical *S. maltophilia* strains, including 23 ocular isolates and one cystic fibrosis isolate, from the culture collection of the School of Optometry and Vision Science at the University of New South Wales, Sydney, Australia, were used to test the lytic range of phage Ps15 (Table 1). The ocular isolates had been collected from eye swabs, contact lenses and contact lens cases from patients with keratitis and from non-keratitis contact lens wearers in the period between February 1994 and February 2012. These strains had been previously investigated for susceptibility to various antibiotic classes and contact lens multi-purpose disinfectant solutions [31]. They were cultivated on Nutrient Agar (NA) (CM0003; Oxoid Australia) or TSA at 37 °C for 24–48 h. Stock cultures were made in TSB with 15% (*w/v*) glycerol and stored at −80 °C.

*P. aeruginosa* PAO1 was obtained from the bacterial culture collection stock of the Centre for Marine Science and Innovation at UNSW and similarly cultured and stored.

### 2.2. Bacteriophage

The lysate of the phage Ps15 was supplied by AusPhage. This phage was isolated from the Cleveland Bay Water Processing Plant in Townsville, where the wastewater from public hospitals and most of Townsville city is processed. The phage was isolated and propagated on the *Pseudomonas aeruginosa* AP143 using standard phage isolation methods [33].

New phage Ps15 stock was prepared from a single purified plaque of the provided lysate. The purification was performed by plaque re-streaking three times in 0.7% (*w/v*) agar (Kobe agar-#44305; Langdon Ingredients, Derrimut, Victoria, Australia) overlay with a sterile platinum loop followed by amplification on AP143S in TSB in the presence of 10 mM CaCl_2_. The lysate was filtered through a 0.22 µm sterile Millipore filter (SLGV033RS; Merck Australia (Bayswater, Victoria, Australia) and stocked with 15% (*v/v*) glycerol at −20 °C. It was given the designation vB_SmaM_Ps15 according to the recommended guidelines for bacteriophage naming by Kropinski et al. [34] (further in the text referred to as Ps15). From the original Ps15 cryotube, 500 µL was concentrated and used in electron microscopy, while all other experiments were performed using the re-amplified phage stock.

### 2.3. Ps15 Titering and Plaque Morphology

Phage titre and morphology were determined using the propagating host *S. maltophilia* AP143S following the standard double-layer agar method [35] on both NA and TSA media, with and without the addition of 10 mM CaCl_2_, and was repeated twice.

### 2.4. S. maltophilia Bacterial Strain Fingerprinting

To investigate the genotypic diversity of *S. maltophilia* strains, Repetitive Polymerase Chain Reaction (rep-PCR) was performed using BOXA1R (5′-CTACGGCAAGGCGACGCTGACG-3′) [36] as a single primer with the PCR cycling conditions of: 2 min at 92 °C, 35 cycles of: 30 sec at 92 °C, 1 min at 40 °C, 2 min at 72 °C; 5 min at 72 °C. Colony-PCR reactions were performed in GoTaq^®^ Green Master Mix (M7122, Promega Australia, Alexandria, NSW, Australia) following the manufacturer’s standard application protocol for a 25 µL volume reaction. The negative control reaction had water substituted for the template DNA. The amplification products (5 µL) were electrophoresed on 1.5% (*w/v*) molecular grade agarose (#1613101; Bio-Rad Australia, Gladesville, NSW, Australia) gels in 0.5x TBE (Tris-borate-EDTA, pH 8.0) at a constant 80 V for 75 min and stained with GelRed^®^ 10,000× in water staining solution in water (#41003; Biotium, Fremont, CA, USA) for 30 min. The amplicons were assessed against the molecular size marker HyperLadder^TM^ 1 kb (#BIO-33053; Bioline Australia, Eveleigh, NSW, Australia) and visually compared.

### 2.5. Host Range Analysis and the Efficiency of Plating (EOP)

The host range of phage Ps15 was tested by spotting 10 μL of serial dilutions of the phage lysate (2.1 × 10^8^ plaque-forming units (PFU)/mL) onto lawns of freshly grown bacteria in 0.7% (*w/v*) top layer NA. The plaques produced on susceptible hosts in the presence of calcium ions were counted and plaque morphology was observed following an overnight plate incubation at 37 °C. Ps15 was screened against a total of 24 *S. maltophilia* strains and *P. aeruginosa* PAO1. The relative EOP was expressed as the ratio between the phage titre of the tested strains and the titre of the propagating host strain [37]. The experiment was performed twice and the resulting EOP represented the mean value.

### 2.6. Transmission Electron Microscopy (TEM)

The particles from 500 μL from the original cryotube of Ps15 phage lysate were concentrated with 10% (*w/v*) PEG8000 in 0.5 M NaCl overnight at 4 °C followed by centrifugation at 22,000× *g* for 30 min. The pellet was washed twice with and then resuspended in 60 µL 0.9% (*w/v*) NaCl. A drop (10 µL) was loaded onto a freshly glow-charged formvar/pioloform carbon-coated 400 mesh copper grid for 2 min to allow time for the phage to absorb. Negative staining was performed with 2% (*w/v*) aqueous uranyl acetate (UA). Phage morphology was examined using a JEM-1400 transmission electron microscope (JEOL Ltd., Tokyo, Japan) and the micrographs were recorded using a Matataki Flash camera and iTEM software (Olympus Soft Imaging Solutions, Münster, Germany). Phage dimensions were calculated from measurements of 15 individual virions using ImageJ software [38].

### 2.7. Phage Assays

The one step growth curve, the burst size and the latent period of the phage Ps15 were determined following a previously described method [39]. The host strain *S. maltophilia* AP143S was grown in the presence of calcium chloride to a mid-log phase and the added phage (10^5^ PFU/mL) was allowed to absorb to cells for 5 min before the sampling commenced. Plaques from each sampling point were counted in a molten overlay and the data were plotted against time. The adsorption assay was performed according to the method of [35]. In brief, tubes with 950 µL TSB containing two drops of chloroform were prepared and placed on ice for 10 min. The bacterial host culture, which was grown to OD_650_ 0.15 in TSB with 10 mM CaCl_2_ (10 mL) (Flask A) was placed in a water bath at 37 °C for 10 min alongside a tube with the same volume of TSB only (Flask C). Phage (1 mL) prewarmed to 37 °C was added at time zero to both flasks at a titre of 1 × 10^5^. At 1 min intervals, 50 µL aliquots were withdrawn from Flask A into the ice-cold tubes and phage was titred by plaque assay. Phage assays were performed on three separate occasions.

Infection growth curves were prepared in 96-well tissue culture plates (Sarstedt, Germany) using the EnSight^TM^ Multimode Plate Reader (PerkinElmer Pty Ltd., Victoria, Australia) according to the protocol described in Leskinen et al. [40]. The lytic ability of the phage Ps15 was tested in a total volume of 200 µL in presence of 10 mM CaCl_2_ by mixing a bacterial culture AP143S at optical density (OD) at 595 nm (OD_595_) of 0.1 and phage at different dilutions in TSB to achieve multiplicity of infection ratios (MOI) of 0.001, 0.01, 0.1, 1, 10 and 100. The OD_595_ was measured every hour for 8 h. To investigate the influence of calcium ions on the cell infection process, lysis of AP143S was monitored on the same plate in the absence of calcium by infecting the culture at the MOIs of 1 and 0.1. The experiment was performed twice in triplicate parallel wells and the mean values were calculated.

### 2.8. Phage Purification and DNA Isolation

Crude phage lysate (1.5 mL) was treated with 10 µg/mL RNase A and 1 µg/mL DNAse I (Sigma-Aldrich Australia, Castle Hill, NSW, Australia) final concentration for 30 min at 37 °C to remove bacterial nucleic acids. Following centrifugation at 22,000× *g* for 10 min to remove debris, phage particles in the supernatant were precipitated with 10% (*w/v*) PEG8000 in 0.5 M NaCl and further treated with proteinase K and sodium dodecyl sulphate as described earlier [41]. Phage genomic DNA was extracted sequentially with phenol, phenol-chloroform-isoamyl alcohol (25:24:1) and chloroform-isoamyl alcohol (24:1; Sigma-Aldrich Australia) with vigorous mixing and centrifugation for 5 min between each step. The final upper aqueous phase was precipitated with 2.5 volumes of ice-cold absolute ethanol, 0.1 volume of 8 M potassium acetate (pH 4.8) and 1 µL of glycogen overnight at −20 °C. After centrifugation at 22,000× *g* for 20 min at 4 °C, the pellet was twice washed with 70% (*v/v*) ethanol and resuspended in 50 μL of TE buffer (pH 8.0). The purity of the isolated DNA was checked by NanoDrop 1000 spectrophotometer (Thermo Fisher Scientific, North Ryde, NSW, Australia) and the concentration measured using Qubit 4 Fluorometer (Thermo Fisher Scientific) with the DNA integrity confirmed on an agarose gel prior to submission for whole genome sequencing.

### 2.9. Illumina MiSeq Phage Ps15 Sequencing and Genome Analysis

The sequencing was performed in the Ramaciotti Centre for Genomics, UNSW, Sydney using a MiSeq system (Illumina; 1× MiSeq reagent kit v2, 2 × 150 bp Nano Sequencing Run). The sequencing library was prepared with 10× Nextera XT DNA Library preparation kit (Illumina). Following quality control (>80% bases higher than Q30), the raw reads were pre-processed with VICUNA v1.0 [42] and assembled with Unicycler v0.4.8 [43].

Gene-calling was performed with multiPhATE v2.0 [44] in ‘consensus’ mode using Prodigal v2.6.3 [45], GLIMMER 3.02 [46], and PHANOTATE v2019.08.09 [47] as gene callers. Gene annotation was obtained by blasting against the NCBI virus genome database, virus orthologous groups (VOG), prokaryotic virus orthologous groups (pVOG), the PhAnToMe database, SwissProt and custom databases composed of all publicly available *Menderavirus* genomes from NCBI (accessed on 21 September 2021, see list below). The search for tRNA genes was conducted with tRNAscan-SE v.2.0.7. Further annotations were provided by searching VOG and pVOG HMM profiles with hmmscan v3.3.1 [48].

Additional annotation methods were used for proteins with hypothetical or unclear annotations. Ortholog annotation was performed with eggNOG-mapper v2.1.2 [49] using eggNOG v5 [50] as the reference. Predicted proteins were also searched against the Conserved Domain Database (CDD) v3.19 [51]. Potential antibiotic resistance genes were searched with AMRFinderPlus v3.10.5 [52].

Promoters were predicted with PePPER [53] on the Genome2D webserver [54]. Rho-independent terminators were predicted with TransTermHP [55] through the Genome2D portal [54]. In case of multiple predictions overlapping the same coordinates and strand, a single prediction was chosen based on confidence value > hairpin score > tail score.

In addition to the experimental evidence, phage lifestyle was predicted with BACPHLIP v0.9.6 [56]. GC and AT skewness were calculated with Genskew online (https://genskew.csb.univie.ac.at/, accessed on 18 March 2022).

### 2.10. Comparative Genomics and Pangenome

The genome of Ps15 was compared with all the publicly available *Menderavirus* genomes: Mendera (type, GCA_009388065.1) IME-SM1 (GCA_002606605.1), Moby (GCA_009388225.1), YB07 (GCA_004521575.1), BUCT608 (GCA_019465995.1) and Marzo (MZ326868.1). To provide a less biased comparison between *Menderavirus* genomes, gene calling for all reference genomes was rerun as indicated above.

Genome visualisation and comparison was performed with GView [57] public server in “BLAST Atlas” mode, using BLASTn on CDS and tRNA sequences using an e-value cutoff of 1e−10, low complexity filter, minimum identity of 70% and minimum fragment size of 50.

Pangenomic analysis was performed with Roary v3.13.0 [58] with an identity threshold of 70% to account for the large variability in viral gene sequences.

### 2.11. Phylogenetics

Protein coding genes present in all genomes (188) were aligned with MAFFT-L-INS-i v7.475 [59] and concatenated. A maximum likelihood tree was constructed with IQ-TREE v2.1.4-beta [60] with 1000 ultrafast bootstrap replicates [61] and automated substitution model selection for each partition. Due to the limited number of conserved genes with the closest non-*Menderavirus* genomes suitable to be used as root, i.e., *Acidovorax* phage ACP17 (GCF_002625205.1), the long evolutionary distance and the high sequence conservancy of the shared genes, the phylogenomic tree was rooted using non-reversible substitution models [62].

Single protein trees were built using the top 100 UniProtKB matches against the respective Ps15 proteins (e-value 0.0001, low complexity regions filter). Matches were dereplicated using CD-Hit v4.8.1 [63] with the following parameters -s 0.95 -c 0.95. With the redundant sequences removed, all the corresponding *Menderavirus* sequences were added. Alignment was performed with MAFFT-L-INS-i v7.475 [59]. Trees were built with IQ-TREE v2.1.3 [60] under the recommended substitution model (-m MFP), with 1000 ultrafast bootstrap replicates and nearest neighbour interchange optimisation [61]. Rooting of the phylogenetic trees was evaluated using non-reversible substitution models [62] using IQ-TREE v2.1.3 [60] with 1000 ultrafast bootstrap replicates (referred to as rootstrapping) [62].

## 3. Results

### 3.1. Ps15 Plaque Morphology

Ps15 formed a typical virulent phage morphology of clear plaques when spotted on its primary host AP143S. The plaques had a slightly smaller diameter on nutrient agar (1–1.5 mm) than on tryptone soya agar (2 mm). The inclusion of calcium chloride in the growth media was not a requirement for plaque formation but did result in a higher lysate titre (2.3 × 10^8^ PFU/mL with vs. 1.2 × 10^8^ PFU/mL without calcium). In the absence of calcium ions, halos were formed around the clear plaque centres (Figure 1b).

During spotting of phage dilutions of 10^8^ PFU/mL lysate onto the lawns of sensitive *S. maltophilia* strains, a diffuse web of bacterial growth was evident on native spots containing high virion concentrations (Figure 1c,d). This suggested a potential lysogeny or a generation of resistant colony growth.

### 3.2. S. maltophilia Strain Fingerprinting

Two *S. maltophilia* isolates, Smal18 and Smal19, displayed an indistinguishable BOXA1R-PCR profile, with Smal7 being very similar, while other isolates had distinctive unique profiles (Appendix A).

### 3.3. Host Range Analysis

Phage Ps15 infected 21 out of 23 *S. maltophilia* ocular isolates and the one isolate from cystic fibrosis (Table 2). Its lytic efficiencies varied from 0.04 to 1.9. The efficiency of plating (EOP) was highest on Smal15 and 16, followed by Smal20 and Xmal2. Another nine strains, Xmal10, Smal6, 8, 9, 10, 11, 14, 17 and 21 also displayed efficient plaquing (low to high 10^7^ PFU/mL), followed by Xmal15 and Smal1 (10^6^ PFU/mL). Smal18 and 19 presented plaques on 10^8^ to 10^5^ PFU/mL, but no single plaques were visible. Two strains, Xmal12 and Xmal21 were phage resistant, as was the strain *P. aeruginosa* PAO1.

### 3.4. Transmission Electron Microscopy

The Ps15 phage virions had isometric heads of 85 ± 2 nm in diameter and straight, non-flexible, striated tails that ranged in length from 93–118 nm (average 110 nm) and in width from 14–21 nm (average 17 nm) (*n* = 15) (Figure 2a–c and Appendix A). Such morphological characteristics were indicative of the *Myoviridae* phage family. The tails ended with plates (31 × 19 nm) and short fibres were visible (Figure 2a). The neck structure had an average length of 12.5 nm and width of 7 nm. Other morphological characteristics that were seen on numerous particles included pearl-like extensions, presumably protein subunits, mostly located at the top of tails just below the neck (Figure 2b) or wrapped around the tails (Appendix A). On some virions, these extensions appeared to emanate from heads at budding sections (Figure 2c). Their lengths were 135–140 nm. Similar structures were previously seen for another *Myoviridae* phage, *S. maltophilia* S3, of a much smaller estimated genome size (~33 kbp) than Ps15 [64].

In addition to the presence of myovirus-type virions, the original Ps15 lysate preparation contained filamentous structures at approximately equal ratios (Appendix A). These filaments appeared flexible and ended with a round or a cylindrical structure on one pole (Figure 3a–d). Their length was an average of 1.25 µm (900–1500 nm) with a width of 10.5 nm (*n* = 9).

### 3.5. Phage Assays

One-step growth curve was determined with phage infection at a MOI 0.001. A latent period of ~40 min and the burst size of 52 ± 5 virions per infected cell represented the average values obtained in three replicate experiments (Figure 4a). The adsorption assay revealed that ~95% of the phages adsorbed to the host cells within 5 min and 98–99% within 10 min (Figure 4b). The adsorption rate constant (*k*) of Ps15 was 9.13 × 10^−8^ mL/min.

The lytic ability of Ps15 against *S. maltophilia* AP143S was investigated in liquid medium at MOI values ranging from 0.0001 to 100 (Figure 5). As expected, the higher the MOI the higher the inhibition of bacterial growth.

Phage Ps15 infection of AP143S appeared to proceed at a faster rate in the presence of calcium ions, yet the overall lysis efficiencies such as evidenced at the start of lysis for MOI 0.1 (Figure 6) were not calcium dependant indicating that Ps15 does not require calcium ions to complete the lytic cycle.

### 3.6. General Genome Characteristics

Assembly of the Ps15 genome resulted in a single circular chromosome of 161,350 bp with a mean coverage of 112× (median 107×) and a GC content of 54.2% (Table 3).

Consensus gene calling predicted 276 protein coding genes and 24 tRNA, including one pseudo tRNA, with specificities for all proteinogenic amino acids (Appendix A). Functional gene predictions enabled putative functional assignments for 95 gene products and the remainder were annotated as hypothetical proteins. The most frequently used start codon was ATG (249 ORFs), followed by GTG (24 ORFs) and TTG (3 ORFs), while TGA was the most frequent stop codon (TGA, 65.58%; TAA, 33.33%; TAG, 0.72% and CCT, 0.36%). The length of the predicted proteins ranged from 62 (ORF35) to 1931 (ORF93) amino acids.

The genome had a quasi-modular organisation (Figure 7), with most of the DNA/RNA processing genes clustered together (66,523–82,903 bp, ORF111–132) after the group of structural genes (39,962–65,295 bp, ORF88–108), while metabolic genes were dispersed through the genome rarely in groups larger than two. A minority of the few protein coding genes were transcribed from a reverse strand cluster together at 24,813–39,588 bp (ORF33–87). The proteins encoded in this reverse strand region were mostly hypothetical and unique to Ps15. A total of 52 promoters (Appendix A) and 33 rho-independent terminators (Appendix A) were predicted, including 6 tail-to-tail terminator regions. As a lysogeny module, integrase or repressor genes were not identified in the genome of Ps15, its lifestyle was predicted as virulent (87.5%). Additionally, no antibiotic resistance gene determinants, toxin or virulent genes were identified.

Most of the tRNA genes (20/24, Appendix A) clustered together within 3.8 kbp (134,989–138,812 bp), while the remaining four were located between 123,174 and 123,825 bp (Figure 7). Both tRNA segments were bound and interspersed with hypothetical proteins, except the gene product Regulatory protein, FmdB family (136,348–136,614 bp), which was located between tRNA-Lys (CTT) and tRNA-Asn (GTT).

Bacteriophage Ps15 shares a high whole genome sequence identity with six other *S. maltophilia* phages (Table 3 and Figure 7) from the recently proposed *Menderavirus* genus (*Myoviridae* < *Caudovirales* < *Caudoviricetes* < *Uroviricota* < *Heunggongvirae* < *Duplodnaviria*) [65]. Based on this high similarity and phylogenetic analysis (Figure 10), Ps15 is a species of the genus *Menderavirus*.

### 3.7. Functional Genome Analysis

Most of the 31 genes encoding proteins for DNA replication, repair and recombination were organised in two subclusters, one at 66,521–82,901 bp (ORF111–132) and the other at 100,399–109,813 bp (ORF158–166). The first subcluster involved the genes associated with DNA synthesis and nucleotide metabolism, such as the recombination endonucleases required for initiation of DNA replication, resolvase, single strand DNA binding protein UvsY, helicase, DNA polymerase and its sliding clump loader subunits. The second subcluster included the late transcription related proteins, such as DNA primase, RNAaseH, helicase-loading protein and two types of exonucleases. RNA ligase (ORF5) and DNA ligase (ORF214) were located outside these subclusters, as was the *pse*T gene, encoding PseT polynucleotide 5′-kinase and 3′-phosphatase (ORF236), involved in the late gene expression and the establishment of the required intracellular DNA structure [66]. DNA packaging enzymes, a small (ORF99) terminase subunit that initiates packaging and a large terminase subunit (ORF100) that translocates DNA, were located within the structural module, between a tail sheath stabiliser and a tail completion protein (ORF98 and ORF101), and adjacent to multiple procapsid assembling proteins (head core scaffolding proteins and a portal head vertex protein). Through the “portal vertex”, DNA is injected into a procapsid powered by ATP hydrolysis and once it is packaged, the pre-assembled tail attaches to it to complete virion assembly [67,68].

The structural module contained 25 structural proteins, which were predominantly located at 39,960–65,293 bp (ORF88–108), while one cluster consisting of five proteins transcribed from both main and opposite strands was distantly located at 5332–8465 bp. Ten structural proteins were baseplate/wedge-linked, eight were associated with tail morphogenesis, including one tail fibre protein and seven with head morphogenesis. The baseplate hub subunit protein ORF20 appeared to also contain a domain involved in host cell lysis, the tail lysozyme. The Ps15 virion structural module was the most conserved T4-like module, as it contained the most gene homologues of T4 and a similar gene order.

There were four putative predicted proteins related to lysis scattered over the Ps15 genome, which include cell wall hydrolase (ORF1, 168 aa), endolysin *N*-acetylmuramidase (ORF116, 157 aa), uncharacterized membrane protein YhgE—Soluble lytic murein transglycosylase (ORF154, 705 aa) (also known in the literature as lysozyme type G (EC 3.2.1.17 and chitinase (EC 3.2.1.14)) and the baseplate hub subunit and tail lysozyme (ORF20, 553 aa). The first two are types of endolysin and the latter two are classified as virion-associated lysins (VALs) as they often form the structural part of tail fibres, tape measure proteins and baseplates. VALs act at the stage of murein degradation from outside the cell and are commonly of larger size and more diverse than endolysins [69].

The metabolic genes of Ps15 were scattered throughout the genome and involved those that encode enzymes for auxiliary metabolism, which are commonly present in genomes of coliphage T4 and T4-related phages [70], such as deoxynucleoside monophosphate kinase (ORF115), DexA exonuclease A (ORF166), ribonucleotide diphosphate reductases NrdA and NrdB (ORF203 and 204), glutaredoxin (ORF206) and thymidylate synthase (ORF217) and in energy metabolism the NAD^+^ biosynthetic pathway, such as nicotinate phosphoribosyltransferase (NadV) (ORF173) and bifunctional nicotinamide adenylyltransferase/ADP-ribose pyrophosphatase NadR (NatV) (ORF174). Bioinformatic analysis of the Ps15 genome has also revealed the presence of a cluster of genes that encode for queuosine biosynthesis enzymes in the 83,414–88,940 bp region. This metabolic pathway involves the set of enzymatic reactions, initiated by ribose cleavage by the enzyme GTP cyclohydrolase I type 1 (ORF135) and include QueC (ORF139), QueE (ORF141) and at a distant position QueD (ORF259). The tRNA Q-modification is accomplished by the replacement of the guanine base with queuine in the anticodon loop of tRNAs by queuine tRNA-ribosyltransferase (ORF140). Due to their essential function, tRNA-guanine transglycosylases (TGT) are conserved in all domains of life [71].

### 3.8. Comparative Genome Analysis and Phylogeny

Pangenome analysis with all the available *Menderavirus* genomes resulted in 386 gene clusters (Figure 8). The core genome of *Menderavirus* includes 188 gene clusters with another 52 gene clusters present in 6 out of 7 genomes (28 of which were absent in Ps15). Out of the 92 singleton gene clusters, 51 pertain to Ps15 and all but one of the Ps15 singletons (ORF89) are placed in the ~15 kbp long coding area of the reverse strand (Figure 7 and Figure 9).

A large stretch of the genome (13.5 ± 0.8 kbp) encoding proteins in the reverse strand appears to be a common feature in all *Menderavirus*, with Ps15 having the longest (14,776 bp) (Figure 9). Based on the raw Clinker output, 30 out of the 55 Ps15 proteins in this region had some match in the other *Menderavirus*; thus 25 proteins remained unique to Ps15. However, 20 out of those 30 proteins had identity values of 30–50%. Similar numbers were obtained with Roary using an identity threshold of 30%. In this case, the number of unique gene clusters was 29/55. Only 3 genes were identical (ORF35, ORF37 and ORF72 in Ps15) and part of the *Menderavirus* core genome (Figure 9). The GC% of this region in Ps15 (55.44%) is similar to the rest of its genome (54.18%). However, the GC skew, and more clearly, the AT skew, show an inversion in the skewness values (positive to negative values and of similar magnitude) (Appendix A).

BLASTp matches of Ps15’s major capsid protein and large terminase subunit identified known *Menderavirus* as top matches (100% id) followed by *Acidovorax* phage ACP17, and with much lower identities, a number of different *Myoviridae* genera infecting Proteobacteria and Cyanobacteria. This is concordant with the phylogenetic analyses, where only *Acidovorax* phage ACP17 appears as a clear close relative to the *Menderavirus* (Figure 10A,B). Even so, the long branches show that ACP17 is noticeably distant to *Menderavirus*. Within the *Menderavirus* genus, Ps15 is shown as a sister taxon to the “M” branch (Mendera, Moby and Marzo) (Figure 10C). Rooting of the tree with non-reversible substitution models resulted in a well-supported root between the “M” group plus Ps15 clade, and the *Stenotrophomonas virus IMESM1* species (IME-SM1, YB07, and BUCT608), with a rootstrap support of 93.8% (the proportion of rooted bootstrap trees with roots in that branching order) (C).

## 4. Discussion

This study represents an initial characterisation of the potential of phage Ps15 to be used in applications related to *S. maltophilia*-associated infections of concern for eye health. To this end, the lytic efficacy of Ps15 was tested against different human clinical ocular strains isolated from patients’ eyes, contact lenses and contact lens cases over a period of 21 years. High intraspecies genetic diversity of *S. maltophilia* clinical and environmental strains has been reported previously [73,74,75]. It is considered to reflect the good adaptability of this species to very diverse environmental conditions, based on the strain-specific acquisition of genes pertinent to different microniches [75,76], to which transposable phages of *S. maltophilia* also contribute by mediating a horizontal gene transfer between strains [77]. This genetic heterogeneity was also observed among ocular strains in this study using repetitive-PCR fingerprinting, which has been evaluated as suitable and highly congruent to MLST and AFLP typing to elucidate *S. maltophilia* intraspecies diversity [78]. Out of 24 strains, 22 displayed genetically distinct profiles, which suggests independent sources of *S. maltophilia* infection of patient eyes or contact lenses, likely driven by host factors or the micro-environment.

The wide lytic range of Ps15 was shown by its ability to infect 90% of *S. maltophilia* strains irrespective of their involvement with microbial keratitis. The observed difference in lytic efficiencies could be attributed to the individual strain genetic make-up, as well as different phenotypic characteristics, such as in relation to previously determined resistance to various antibiotics and multipurpose disinfectant solutions (MPDS) (Table 1) [31]. Most of the isolates displayed medium to high production efficiency, while three ocular isolates (Xmal15, Smal7 and Smal13) and one cystic fibrosis isolate (Smal1) had low lytic efficiencies. Two genetically identical strains, Smal18 and Smal19, appeared to be susceptible to infection by Ps15 only at high phage titres and Xmal12 and Xmal21 were phage resistant. It is possible that the efficacy and coverage could be improved with the adaptation method (coevolutionary phage training), as has been demonstrated previously for initially intermediate or resistant uropathogenic strains [79].

There was no correlation between the phage susceptibility of ocular isolates and their antibiotic resistance profiles. The only differing resistance patterns were with Xmal21, the only strain resistant to ofloxacin and Smal18 and 19, with resistance to the disinfectant solution Biotrue (BT). Whether the resistance determinants could have interfered with phage adsorption or replication could not be speculated without further testing. *S. maltophilia* is a prototype bacterial species with intrinsic and acquired resistance to antibiotics due to the presence of a plethora of genes encoding efflux pumps, beta-lactamases and aminoglycoside inactivating enzymes and can also acquire a transient phenotypic antibiotic resistance induced by host-related factors [80,81]. Hence, it is of relevance that a capacity of Ps15 for lytic infection of multidrug resistant and disinfectant solution resistant clinical ocular strains of *S. maltophilia* was demonstrated in this study.

Electron microscopy examination of the originally received Ps15 lysate amplified on the mixed strain AP143 (*S. maltophilia* and *P. aeruginosa*) revealed an interesting observation. In addition to the myoviral morphology virions, filamentous structures of ~1 µm with mostly round structures on one end were seen. Without further investigation the nature of these filaments can only be speculated, with two possible assumptions considered: that of an authentic filamentous virus or bacterial flagella. The presentation of different morphological forms has been a long-known feature of some eukaryotic viruses, such as influenza virus, which produce a filamentous type (in addition to a spherical type) suggested to facilitate their spread through mucus [82]. The formation of such pleomorphic particles has not been reported for bacterial viruses. AP143 cells may have been permissive to induction of filamentous prophages, or the filaments could arise as a morphological form akin to the example with influenza. Filamentous phages of the *Inoviridae* family capable of establishing chronic infection cycle and possibly increasing the pathogenicity of its host have been described in *S. maltophilia* [27]. As both *S. maltophilia* and *P. aeruginosa* are motile bacterial species, an alternative explanation could be that the observed filaments were free flagella, which were detached from the cells upon cell lysis and possibly broken in the process of lysate preparation (e.g., by centrifugation) and co-precipitated with phage virions. However, it is generally expected that flagella release would require much more concerted effort than applied in a routine phage lysate preparation. Additionally, no characteristic flagellar hook could be discerned on filaments and flagella filaments are normally much bigger (~45 nm in width and >15 µm long) [83]. Resolving the identity of these filamentous particles in the phage preparation would require their targeted extraction and purification to further examine their composition, the process of their formation and possible relevance.

Currently isolated *S. maltophilia* phages include diverse morphological types [27] and range in size from the smallest filamentous phage ΦSHP2 of 5819 bp [84] to the largest SMA5 estimated to be ~250 kbp [28]. With its size of 161,350 bp, phage Ps15 has been placed in the group consisting of six large *Myoviridae* phages: Mendera [85], Moby [86], IME-SM1, YB07, Marzo and BUCT648 that belong to the new *Menderavirus* genus. Mendera and Moby have been isolated in America, IME-SM1 and YB07 in China and Ps15 in Australia from the same type of environment: wastewater. The global existence of these very similar phages apparently mirrors the phylogeographic distribution of their *S. maltophilia* host strains. A recent large whole genome study of the *S. maltophilia* complex has identified 23 monophyletic, worldwide distributed lineages, with the largest number of strains represented by the particular lineage Sm6 adapted to infect humans and predicted to have high resistance to antibiotics and disinfectants [75]. Another study revealed a clear separation into the phylogenetic cluster of either environmental or human isolates with a smaller number of genogroups present within each cluster [74]. It is not known which lineages the bacterial hosts of *Menderavirus* belong to or if they could be anthropogenically influenced environmental isolates, such as excreted by humans or animals and discharged into the wastewater [78], but the high phage similarity could indicate their possible close genetic relatedness.

Phage Ps15 is a T4-like *Myovirus* that shares essential conserved genes encoding proteins involved in DNA replication, recombination and repair and virion morphogenesis (the capsid and tail genes) with the coliphage T4 [87], but their limited DNA and protein homology indicates its substantial evolutionary distance. Genome analysis of Ps15 and other species of *Menderavirus* revealed certain features they shared with the so-called jumbo phages (genome size of >200 kbp), such as a large number of tRNAs; a number of scattered or sub-clustered genes; longer median lengths of proteins; concanavalin A-like lectin protein associated with surface adhesion of tail fibres; metallophosphoesterase with a predicted role in counter-nucleotide defences; methyltransferases that modify phage DNA to evade host restriction; and the presence of enzymes involved in nicotinamide adenine dinucleotide (NAD^+^) metabolism [88,89]. The detection of *nad*V and *nat*V genes suggests the functional pyridine nucleotide scavenging pathway during the metabolic phase of infection for deoxyribonucleotide synthesis [90]. The genes encoding the bifunctional NatV enzymes of the NAD^+^ metabolism are commonly present in phages that infect hosts in aquatic environments (e.g., *Aeromonas*, *Caulobacter*, and *Vibrio*), which may indicate their origin and evolutionary history [89]. Additionally, poly(ADP-ribose) polymerase catalytic domain protein (ADP ribosyltransferase) (ORF165), the second largest protein in Ps15, is present in large phages [91]. Although ADP ribosyltransferase have been associated with toxicity, new findings from studying commensal gut bacteria suggest that they represent fitness factors that phages transfer to bacteria to support their survival and confer them competitive advantage [92].

In a canonical lytic model, both holins and endolysins are required to accomplish host cell lysis, where holin permeabilises the cytoplasmic membrane to allow endolysin to reach and degrade murein [93]. Another functional group of recently characterised lytic proteins termed spanins, as they are predicted to span the entire periplasm, has been suggested as essential for lysis of Gram-negative hosts [93,94]. This is based on findings that the destruction of peptidoglycan layer is not sufficient to ensure lysis and that the release of progeny virions requires the disruption of their outer membrane, most plausibly by mediating fusion between the inner and outer membranes [94]. A gene encoding a putative holin protein has not been identified in the Ps15 genome; however, a gene encoding a putative lipoprotein (ORF192) could possibly have spanin-like characteristics and form a part of Ps15 lytic complement.

Transfer RNA genes are widespread in different viral families and, in addition to delivering amino acids to the ribosome during protein synthesis, can play virus-specific roles in regulating translation, packaging and priming reverse transcription [95]. Their presence is typically characteristic for large phage genomes of the Caudovirales order, particularly of the *Myoviridae* family that infect Proteobacteria [95,96]. It has been revealed from the study involving a large viral dataset that the number of tRNAs positively correlated with the genome length, but negatively correlated with the tRNA genes clustering and that the majority of phages that carried tRNA gene clusters were virulent [96]. The genome of Ps15 harbours all the universal 20 tRNA isotypes, which are arranged within two separate clusters in 1 to 5 genes with a density of ~5 tRNAs/kbp. Ps15 encodes its own unique tRNA gene complement, with no similarity of these genes to any known sequence in the databases. It is expected that they could contribute to the reduction in host dependence of Ps15, increased virulence and fitness, and extension of its host range, as has been previously suggested [95,96,97].

One noticeable feature of Ps15 and other species of *Menderavirus* is the presence of distinctive queuosine genes in their genomes. Queuosine is a hyper-modified nucleoside derivative that modifies tRNAs specific for four amino acids (Asp, Asn, His, or Tyr) by replacement of guanine at position 34 [98,99]. The *que* genes of *Menderavirus* had only weak homology to bacterial *que* genes, indicating their significant divergence. In bacteria, the modification of tRNAs by the inclusion of queuosine improves the specificity of transcription and codon recognition and is involved in regulation of cellular physiology, control of aerobic and anaerobic metabolism, bacterial virulence, etc. [100]. Genes involved in the biosynthesis of queuosine have also previously been identified in bacteriophages and viral metagenomes, in particular from aquatic environments [101,102,103]. Several steps of the queuosine biosynthesis pathway create 7-deazaguanine derivatives that when present in the phage genomes have been shown to allow the escape from the restriction-modification systems of the bacterial hosts [104].

Apart from authentic phage ORFs, Ps15 genome analysis uncovered putative proteins of presumably bacterial origin, which could have been acquired during viral infection of a host or captured from an organism present in the same environment via a horizontal gene transfer. These bacteria-like proteins include: COG3866 (Pectate lyase; ORF28), SET-domain protein (ORF62), S8 family protease (ORF145), and the myo-inositol-1-phosphate synthase (EC 5.5.1.4; ORF137) with 65% homology over 60% query cover to *Saccharibacteria* bacterium. Furthermore, pyrimidine dimer DNA glycosylase (ORF111) had a BLAST-homology of 45–55% over almost the whole genome length with genes from different species of Proteobacteria and Archaea. The concanavalin A-like lectin/glucanases superfamily protein (ORF30) shared similarities to respective proteins from *S. maltophilia.*

The domains of other proteins with unknown functions in phage Ps15 include: Glycosyl hydrolase 8 (cellulase D) family protein (ORF21); COG3391 (a putative amine dehydrogenase; ORF25); COG3866 (pectate lyase; ORF28); SET domain protein (ORF62); RyR domain containing protein (ORF96); DUF4406 domain-containing protein (ORF212) and DUF932 domain-containing protein (ORF240). The predicted putative amine dehydrogenase and pectate lyase are two enzymes with potential value in biotechnological applications. However, an experimental verification would be required to demonstrate their enzymatic activity. Ryanodine receptors (RyR) domains are highly conserved across all life domains and represent a class of intracellular calcium release channels [105]. The gene encoding the protein with this domain in the Ps15 genome was found between two neck proteins. While the important role of RyR domain proteins for human health has been elucidated [106], their function in bacteriophages has yet to be explained.

Basic leucine zipper (bZIP) domain protein (ORF8) is a transcription factor, which contains a basic DNA binding domain and a dimerization domain, a leucine zipper [107]. bZIP proteins belong to a class of DNA binding proteins well-defined in eukaryotes [108], but also encoded by human viruses to modulate transcription during lytic infection [109]. Both bZIP domain protein and EFG_IV domain protein (Elongation factor G domain IV; ORF170) that mediates translation do not have homologs in the genome database, hence the origin of the genes encoding the proteins with these domains is unknown.

Pangenome analysis of the *Menderavirus* revealed that the largest genomic region showing differences across the genus contained proteins encoded on a reverse strand over 13.5 ± 0.8 kbp. The change in the GC skew (sign change) has been previously reported in bacteria as the signature of genomic inversions [110], and its presence across the genus suggests a genomic inversion ancestral to all *Menderavirus*. Ps15 is markedly different to other species of this genus, with 90.9% of all the genes in this region unique to it. In contrast, 50–60% of the genes in the homologous region are shared among the non-Ps15 *Menderavirus*. Further evaluation of the reverse region using blastp (min. id. 30%) showed that some of the proteins encoded in Ps15’s reverse region were shared with other genomes, albeit with very low identity values, especially when considering the overall high identity of the protein coding genes (e.g., 123 of the 188 core genes are ≥95% identical). These extremely low identity values suggest that Ps15 proteins encoded in the reverse region have higher evolutionary rates and could potentially promote the rise of new genetic variants such as those induced by host immune response during infection or under selective pressure upon host co-infections in natural environment.

Comparative analysis of the major capsid and the large terminase proteins of Ps15, selected as phylogenetic markers, revealed limited homology to the respective proteins of the *Acidovorax* phage ACP17. Other Ps15 proteins that displayed similarity of 44–60% over full genome length to those of ACP17 were also identified, such as baseplate wedge protein subunits, tail tube initiator and tail completion protein, lysis protein VrlC, dexA endonuclease. This distant relationship between phage ACP17, lytic to a plant pathogen *Acidovorax citrulli* that causes fruit blotch, and *Stenotrophomonas* phage IME-SM1 has been uncovered previously [111] and can be considered unusual given their vastly different ecological environments.

## 5. Conclusions

This study presented the preliminary characterisation of the *Stenotrophomonas* phage Ps15, a new species of the recently established *Menderavirus* genus. While more biological characterisation would be required to prove its usefulness for therapeutic applications, such as in relation to its stability under different physiological conditions and the antimicrobial activity against a wider range of relevant clinical strains, this initial study has demonstrated the capability of Ps15 to infect genetically diverse ocular isolates of *S. maltophilia*. In addition, its lack of integrase, antibiotic resistance or toxic genes strengthens its potential therapeutic use in ophthalmic formulations to control *S. maltophilia*-associated eye infections.

## Figures and Tables

**Figure 1 viruses-14-00709-f001:**
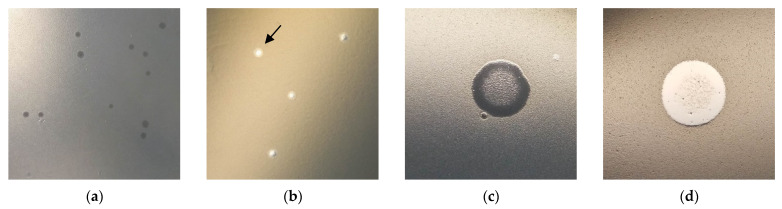
Ps15 plaque morphology on: (**a**) TSA with calcium chloride; (**b**) TSA without calcium chloride (the arrow points to a halo); (**c**) 10 µL spot on *S. maltophilia* Smal2; (**d**) 10 µL spot on *S. maltophilia* Smal17.

**Figure 2 viruses-14-00709-f002:**
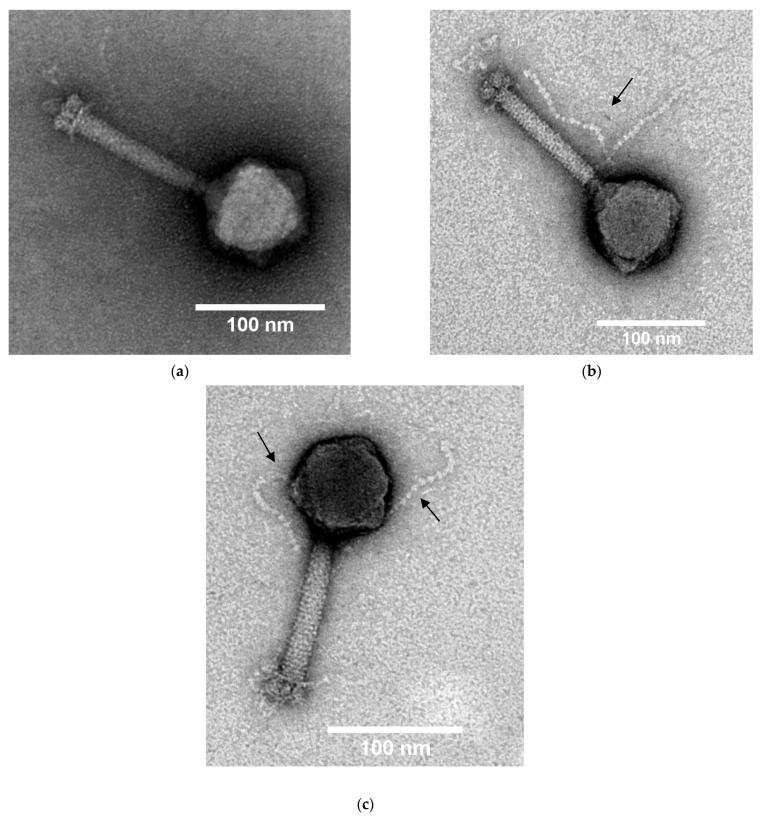
Transmission electron micrographs of the Ps15 phage particles stained with 2% (*w/v*) uranyl acetate displaying the Myoviridae morphotype. Magnification × 60 K. The extensions from a tail (**b**) and a capsid (**c**) are indicated by arrows. Note the specimens with straight short fibres (**a**) and their transformation into globular-looking forms (**b**). Scale bars are included for each image.

**Figure 3 viruses-14-00709-f003:**
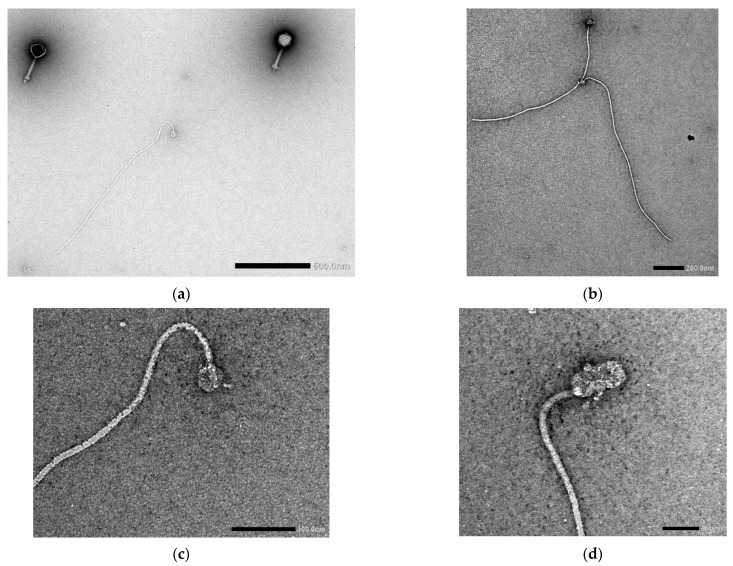
Electron micrographs of 2% (*w/v*) uranyl acetate negatively stained filamentous structures in the original Ps15 phage lysate. Acceleration voltages: (**a**) 15 kV; (**b**) 20 kV; (**c**) 80 kV and (**d**) 100 kV. Scale bars are included in each figure.

**Figure 4 viruses-14-00709-f004:**
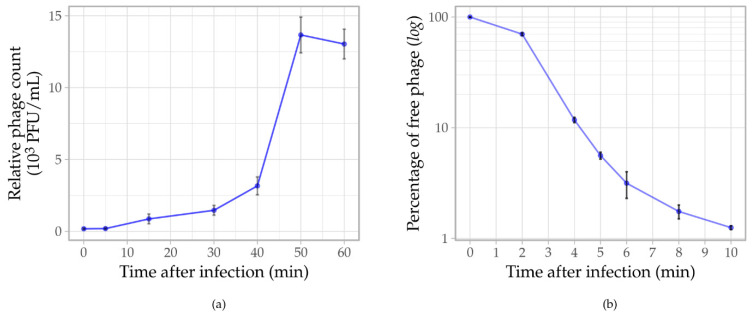
(**a**) One step growth curve depicting the infection of *S. maltophilia* AP143S by phage Ps15 at a MOI of 0.001. (**b**) Phage adsorption assay. Error bars indicate the standard deviation of triplicates.

**Figure 5 viruses-14-00709-f005:**
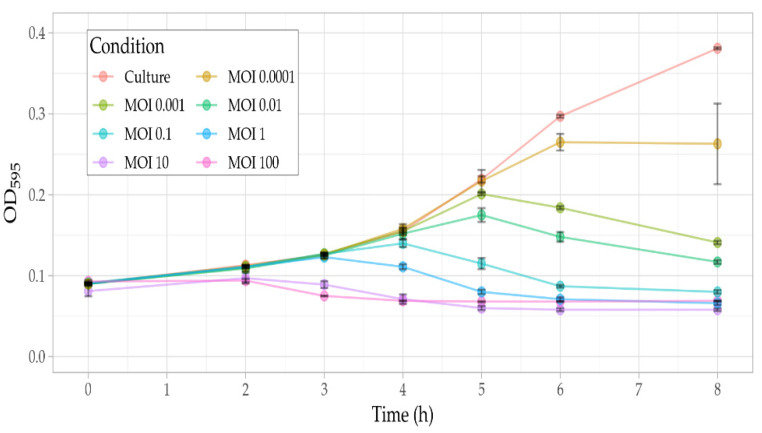
Kill curves of phage Ps15 at different MOIs showing progressive lysis of the host *S. maltophilia* AP143S in TSB at 37 °C. Growth inhibition was assessed against the non-infected bacterial culture used as a control. The plotted values are averaged from six replicate OD_595_ measurements. Error bars indicate the standard deviation of the replicates.

**Figure 6 viruses-14-00709-f006:**
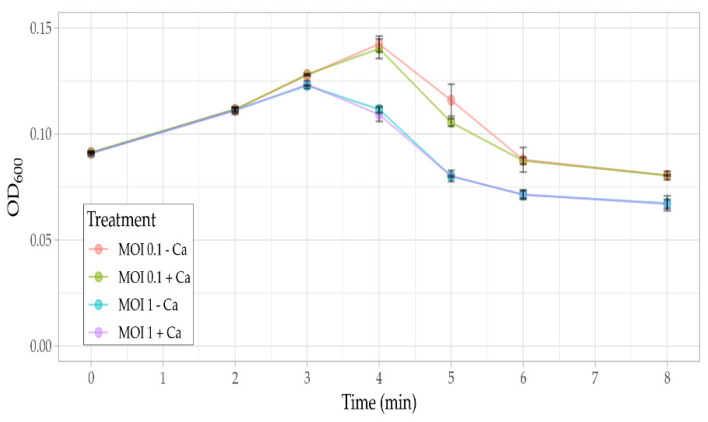
Lysis of *S. maltophilia* AP143S by Ps15 (MOI 0.1) in tryptone soy broth at 37 °C, with (10 mM) and without the addition of calcium chloride ions. Error bars indicate the standard deviation of triplicates.

**Figure 7 viruses-14-00709-f007:**
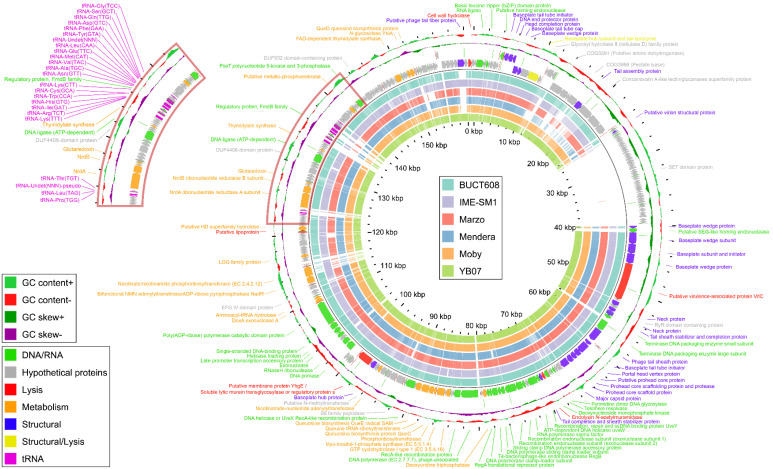
Genome map of Ps15. Outer line graphs depict the GC content and GC skew of the genome. The genome features are coloured based on the gene type, function of the encoded protein, or genomic feature. Inner coloured rings show high identity matches of other *Menderavirus* genomes against Ps15. The red box marks an area of tRNAs with a magnified image in the top left corner showing their position on the genome. Image generated with GView [57].

**Figure 8 viruses-14-00709-f008:**
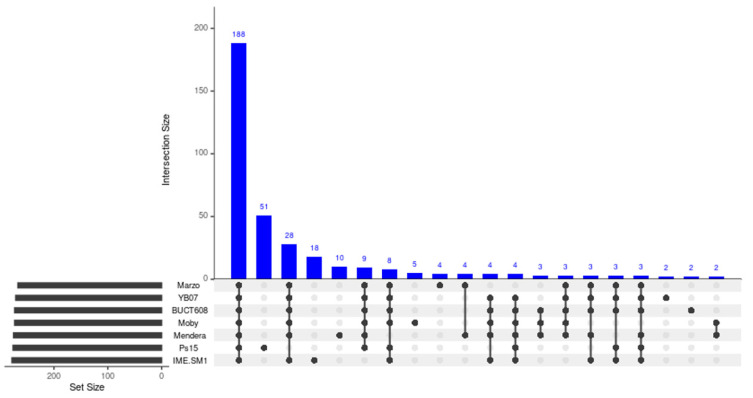
Upset plot depicting the number of shared and unique gene clusters in the *Menderavirus* genomes. Note that after the core genome (188 gene clusters), the Ps15 singletons (51) and the genes only absent in Ps15 (28) constitute the largest intersections.

**Figure 9 viruses-14-00709-f009:**
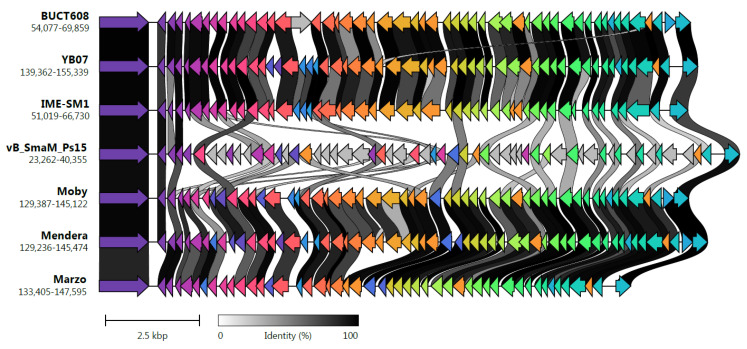
Detailed pairwise comparison of the long cluster of genes encoded in the reverse strand of *Menderavirus*. The view is extended to the first conserved genes in the positive strand at each side of the region. The strains are ordered based on Figure 8C. Gene links are shaded based on identity (minimum 30%). Image generated with Clinker v0.0.21 [72].

**Figure 10 viruses-14-00709-f010:**
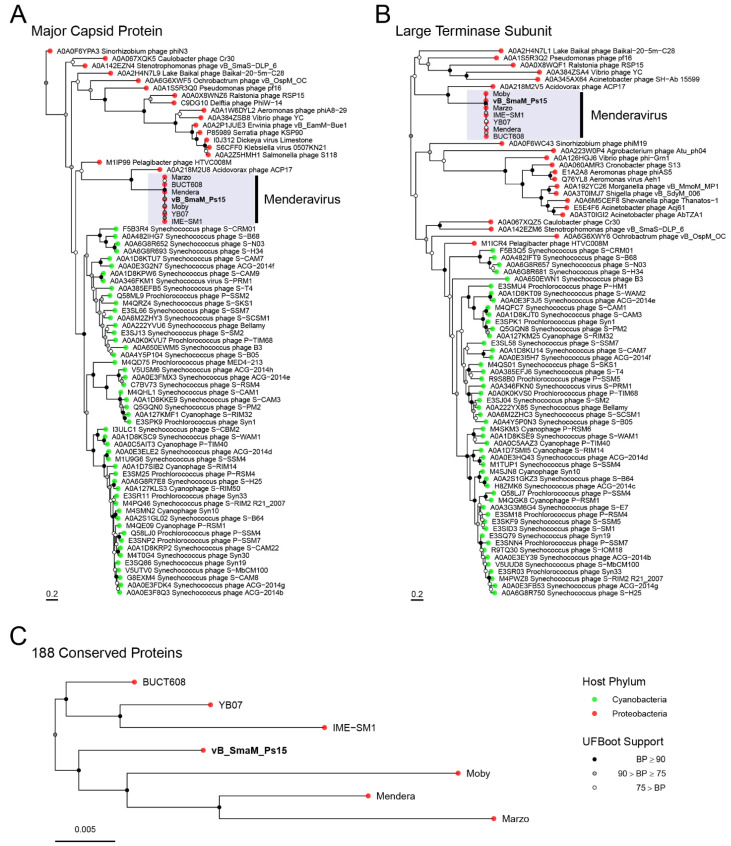
Phylogenetic trees showing the placement of Ps15 in respect to other phages. Phylogenies are based on the major capsid protein (**A**), the large terminase subunit (**B**), and on the partitioned phylogenomic analysis with 188 core proteins (**C**). Tip colours indicate the phylum of the host. Trees were rooted using non-reversible substitution models.

**Table 1 viruses-14-00709-t001:** The *S. maltophilia* strains used for susceptibility testing with phage Ps15.

Strain Name	Isolation Date	Isolation Source	Keratitis-Associated (Y/N)	Antibiotic Resistance *	MPDS Resistance ^#^
Xmal2	4 February 1994	Eye swab	N	azt (I), imi (R), tob (R), chl (I)	AQ, RH, MC
Xmal7	23 September 1994	Eye swab	Y	azt (I), imi (R), chl (I)	AQ, RH, MC
Xmal10	15 December 1994	Eye swab	Y	azt (I), imi (R), chl (I)	AQ, RH, EM, MC
Xmal12	12 October 1995	Eye swab	Y	ND	ND
Xmal15	19 March 1998	Contact lens	Y	azt (I), imi (R), chl (R)	AQ, RH, MC
Xmal21	2 April 2001	Contact lens	Y	cip (R), ofl (I), imi (R), gen (R), tob (R), chl (I)	AQ, RH
Smal1	12 December 2005	Cystic fibrosis swab	N	azt (I), imi (R), chl (I)	AQ, RH, MC
Smal2	25 May 1998	Contact lens case	N	cip (I), imi (R), gen (I), chl (I)	AQ, RH, MC
Smal6	9 September 2010	Contact lens case	Y	azt (R), cep (R), imi (R), chl (I)	AQ, RH, MC
Smal7	11 November 2010	Contact lens case	N	azt (R), imi (R)	AQ, RH
Smal8	26 November 2010	Contact lens case	N	imi (R), chl (I)	AQ, RH
Smal9	1 February 2011	Contact lens case	N	imi (R)	AQ, RH, MC
Smal10	14 February 2011	Contact lens case	N	imi (R)	AQ, RH, MC
Smal11	9 March 2011	Contact lens case	N	azt (I), imi (R), pmb (R)	AQ, RH, EM, MC
Smal12	24 March 2011	Contact lens case	Y	imi (R)	AQ, RH, MC
Smal13	1 April 2011	Contact lens case	N	azt (R), cep (R), imi (R),	AQ, RH, MC
Smal14	5 April 2011	Contact lens case	N	azt (I), imi (R), chl (R),	AQ, RH
Smal15	7 April 2011	Contact lens case	N	azt (R), imi (R), chl (I)	AQ, RH, MC
Smal16	17 May 2011	Contact lens case	N	azt (I), imi (R),	AQ, RH, MC
Smal17	6 June 2011	Contact lens case	N	imi (R)	AQ, RH, MC
Smal18	1 July 2011	Contact lens case	N	azt (R), cep (I), imi (R),	BT, AQ, RH, MC
Smal19	8 July 2011	Contact lens case	N	azt (I), cep (R), imi (R)	BT, AQ, RH, MC
Smal20	5 July 2011	Contact lens case	N	azt (R), cep (I), imi (R), tic (R), chl (I)	AQ, RH, MC
Smal21	February 2012	Contact lens case	N	ND	ND

*, tested against; ciprofloxacin (cip), gatifloxacin (gat), levofloxacin (lev), moxifloxacin (mox), ofloxacin (ofl), aztreonam (azt), ceftazidime (cez), cefepime (cep), imipenem (Imi), ticarcillin (tic), gentamicin (gen), tobramycin (tob), tigecycline (tig), trimethoprim/sulfamethozole (cotrimoxazole; cot), chloramphenicol (chl), polymyxin B (pmb); I = intermediate resistance, R = resistant, ND = not determined. ^#^, resistance to MPDS defined as being growth in >10% *v/v* multipurpose disinfectant solutions (MPDS) [32]; RevitaLens (RL), ReNu fresh (RF), Biotrue (BT), Aquify (AQ), RepleniSH (RH), EverMoist (EM), MeniCare (MC).

**Table 2 viruses-14-00709-t002:** Efficiency of plating (EOP) of phage Ps15 on different bacterial hosts expressed as the ratio of the phage titre of the tested strains and the titre of the propagating host strain (2.1 × 10^8^ PFU/mL).

Strain Name	Relative Efficiency of Plating	Single Plaque Description
Smal15	1.9	clear
Smal16	1.8	clear
Smal20	1.67	clear
Xmal2	1.17	clear
Smal2	1.08	turbid
Xmal10	1	clear
AP143S	1	clear
Smal12	0.91	clear
Smal6	0.82	clear
Smal11	0.8	clear
Smal21	0.76	clear
Xmal7	0.75	turbid
Smal14	0.42	turbid
Smal17	0.38	clear
Smal8	0.28	clear
Smal9	0.23	clear
Smal10	0.11	clear
Xmal15	0.06	clear
Smal1	0.04	clear
Smal7 *	0.04	turbid
Smal13 *	0.03	turbid
Smal18	No single plaques (plaque spots at 10^5^–10^8^ PFU/mL)	-
Smal19	No single plaques (plaque spots at 10^5^–10^8^ PFU/mL)	-
Xmal12	-	-
Xmal21	-	-
PAO1	-	-

* Smal7 and Smal13 produced no lysis in the initial test and single plaques at the level 10^5^ PFU/mL in the repeated test. ‘-’—no plaque.

**Table 3 viruses-14-00709-t003:** General characteristics of Ps15 and other known *Menderavirus* genomes. Values based on the annotations performed in this study unless otherwise stated.

Phage Name	Size (bp)	GC%	Proteins ^a^	Coding Density	tRNAs ^ab^	Genome Similarity/Query Cover (%) ^c^	Average Amino Acid Identity (%) ^d^	Shared Proteins (%)
Ps15	161,350	54.2	275	93.86	24	100.0/100	100	276 (100)
BUCT608	160,122	54.2	270 (266)	94.15	24 (20)	98.62/92	93.90	221 (81.85)
IME-SM1	159,514	54.1	267 (202)	92.88	24 (20)	98.50/91	93.63	210 (78.65)
Marzo	159,384	54.0	262 (268)	93.65	24 (23)	97.49/90.5	91.75	213 (81.30)
Mendera	159,961	54.0	272 (286)	94.05	24 (23)	98.07/89.5	91.13	207 (76.10)
Moby	159,365	54.1	268 (271)	94.02	24 (23)	94.73/91.5	91.98	215 (80.22)
YB07	159,862	54.1	269 (257)	93.85	24 (0)	98.45/90.5	93.47	220 (81.78)

^a^: values from NCBI’s public genome annotations in parenthesis. ^b^: includes predicted pseudo-genes and tRNAs of indetermined specificity. ^c^: based on genome vs. genome with blastn. The values indicate the average of reciprocal hits. ^d^: see Appendix A for full analysis.

## Data Availability

The genome sequence of vB_SmaM_Ps15 was deposited at NCBI under GenBank accession number OL702939.

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
