# Peer review of "Characterisation of Bacteriophage vB_SmaM_Ps15 Infective to *Stenotrophomonas maltophilia* Clinical Ocular Isolates"

_viruses, 2022, doi:10.3390/v14040709_

Round 1
Reviewer 1 Report
Damnjanović and coworkers characterize a novel bacteriophage vB_SmaM_Ps15 using morphological analysis, host infectivity assays, and comparative genomics analysis techniques. Based on their data, the authors propose that this novel bacteriophage might be used as a phage therapy for treatment of S. maltophilia eye infections although further studies would be needed to substantiate this claim. Overall, the manuscript is well written and provides a thorough characterization of a novel bacteriophage and its potential utility to the research community.
Minor Concerns:
- Lines 36-37 – list of isolation sources should be clarified, or commas added to increase readability.
- Line 51 – Paragraph starts of describing microbial keratitis and quickly changes to a description of P. aeruginosa isolates from ocular infections. It is not clear the link between the two sentences. An additional sentence would aid the readers and improve clarity.
- Table 2 – Table should be sorted by strain name or relative efficiency of plating. Sorting by relative efficiency of plating would allow readers to quickly identify the strains that were most susceptible to infection.
- Line 490 – “rootstrap support” it is not clear what rootstrap support is. Do the authors mean “bootstrap support”?
Author Response
Please see attached Word document

Reviewer 2 Report
This article presents the isolation, sequencing, genomic analysis, one-step growth curve and clinical host range analysis of phage vB_SmaM_Ps15. This analysis may be useful in the treatment of S. maltophilia infections.
Minor suggestions:
1) line 284 Please reference BOXA1R-PCR (line 284, reference[36] ?). Also, BOXA1R-PCR is written differently in the methods and results. Should it be BOXA1R-PCR or BOXA1-R PCR?
2) line 289: "another seven strains" should be "another nine strains"?
3)line 331: "UA" should be Uranyl acetate?
4) Figure 7: Please reference the software used to make the figure in the figure legend (GCView?)
5) The region from 24,813 - 39,588 bp (ORF33 - 87) is quite interesting. Does it harbor a different GC content than the rest of the genome, suggesting it is a more recent acquisition? It appears to in Figure 7. Please at GC content details to the discussion of this region
6) line 650: it is curious you consider a capsid decoration protein and putative tail fiber protein (line 655) of bacterial origin. If these have BLAST hits to bacteria, it is likely to temperate phages integrated in bacterial lysogens.
Author Response
please see attached word document
